# Characterization of *Plocamium telfairiae* Extract-Functionalized Au Nanostructures and Their Anti-Adipogenic Activity through PLD1

**DOI:** 10.3390/md20070421

**Published:** 2022-06-27

**Authors:** Sun Young Park, Hye mi Kang, Woo Chang Song, Jin-Woo Oh, Geuntae Park, Young-Whan Choi

**Affiliations:** 1Bio-IT Fusion Technology Research Institute, Pusan National University, Busan 46241, Korea; 2Department of Horticultural Bioscience, Pusan National University, Myrang 50463, Korea; mimi2965@naver.com; 3Department of Nanofusion Technology, Pusan National University, Busan 46241, Korea; dck3202@naver.com (W.C.S.); ojw@pusan.ac.kr (J.-W.O.); gtpark@pusan.ac.kr (G.P.)

**Keywords:** *Plocamium telfairiae*, PT-AuNS, adipogenesis, PLD1

## Abstract

Here, Au nanostructure (AuNS) biosynthesis was mediated through ethanolic extract of *Plocamium telfairiae* (PT) without the use of stabilizers or surfactants. PT-functionalized AuNSs (PT-AuNSs) were analyzed using ultraviolet–visible spectroscopy, dynamic light scattering, high-resolution transmission electron microscopy, energy-dispersive spectroscopy, and Fourier-transform infrared spectroscopy. Stable monodisperse PT-AuNSs were synthesized, with a mean size of 15.36 ± 0.10 nm and zeta potential of −35.85 ± 1.36 mV. Moreover, biosynthetic AuNPs with a face-centered structure of PT-AuNS exhibited crystalline characteristics. In addition, many functional groups playing important roles in the biological reduction of PT extracts were adsorbed on the surface of PT-AuNSs. Furthermore, the effects of PT-AuNSs on adipogenesis in immature adipocytes were investigated. PT-AuNSs reduced morphological changes, lowered triglyceride content, and increased lipid accumulation by approximately 78.6% in immature adipocytes compared with the values in mature adipocytes (MDI-induced). PT-AuNS suppressed lipid accumulation by downregulating the transcript and protein expression of C/EBPα, PPARγ, SREBP 1, FAS, and aP2. Finally, PT-AuNS induced the transcript and protein expression of UCP1, PRDM16, and PGC1a, thereby increasing mitochondrial biogenesis in mature adipocytes and effectively inducing brown adipogenesis. In this study, the biosynthesized PT-AuNS was used as a potential therapeutic candidate because it conferred a potent anti-lipogenic effect. As a result, it can be used in various scientific fields such as medicine and the environment.

## 1. Introduction

According to WHO (World Health Organization) statistics for 2021, 13% of adults worldwide are obese and 39% of adults worldwide are overweight. In particular, 1 in every 5 adolescents and children worldwide is overweight [1]. Obesity is a metabolic disorder caused by the excessive accumulation of fat due to the proliferation and differentiation of preadipocytes in the body. In addition to causing mild chronic inflammatory conditions, obesity can lead to complications such as metabolic diseases, including insulin resistance syndrome, cardiovascular disease, respiratory disease, osteoarthritis, type 2 diabetes, and prostate and breast cancer [2,3,4]. Therefore, the management and treatment of obesity are meaningful in the context of not only obesity itself, but also the aforementioned complications. Energy in the body is stored in the form of triglycerides in fat cells. When the energy reserves in the body deplete, the stored fat is degraded to free fatty acids and glycerol for use as energy sources [5,6]. Therefore, excessive energy intake promotes the differentiation of preadipocytes and increases the amount of lipids stored in the body, which is a direct cause of obesity [7,8]. Adipocytes and adipose tissue, which play key roles in obesity, are crucial for homeostasis in metabolic processes, such as angiogenesis, immune response, steroid metabolism, and hemostasis. However, when the amount of energy absorbed is higher than the amount consumed, the number and volume of fat cells increase, thereby increasing adipose tissue mass [9]. Adipose tissue mass increases through the increase in the number and size of adipocytes. Typically, the size of adipocytes can be controlled through dietary regulation; however, controlling the differentiation of new precursor adipocytes into adipocytes is ineffective through dietary regulation [2]. Therefore, for the fundamental treatment or suppression of obesity, adipocyte differentiation must be controlled efficiently [10,11].

Mature adipocytes are produced through preadipocyte differentiation, which is initiated by hormones and cell differentiation-inducing substances, such as 3-isobutyl-1-methylxanthine, dexamethasone, and insulin (MDI) [12]. Signal transduction through differentiation-inducing substances and insulin activates the phosphatidylinositol 3-kinase (PI3K)/protein kinase B (AKT) pathway in preadipocytes to induce their differentiation into mature adipocytes and produce adipogenesis signals, which further promotes adipogenic metabolism through the increase in cellular glucose uptake [13,14]. As adipocytes that are morphologically similar to fibroblasts become round, characteristics accompanying the mutation of mature adipocytes appear gradually [15]. Through this process, lipid droplets which appear in the adipose cytoplasm grow and merge through differentiation to form multiple lipid droplets. During adipocyte differentiation, triglycerides are accumulated, and the expression of adipocyte-specific proteins—such as fatty acid synthase (FAS) and adipocyte fatty acid-binding protein (aP2)—is induced. The transcription factors peroxisome proliferator-activated receptorγ (PPARγ), CCAAT-enhancer-binding protein α (C/EBPα), and adipocyte determination- and differentiation-dependent factor 1 (ADD1)/sterol regulatory element-binding protein-1c (SREBP1c) play key roles as upstream factors, inducing the expression of these proteins and promoting direct triglyceride production by transporting free fatty acids into the cytoplasm [16,17,18]. Mouse embryonic fibroblasts (3T3-L1) are a widely used cell line in biological studies on adipose tissues. When 3T3-L1 cells are treated with cell differentiation-inducing agents, such as MDI, the lipogenesis signaling pathway is activated, the amount of intracellular fat increases, and the cell shape changes to circular [19,20]. Therefore, candidate substances inhibiting preadipocyte differentiation, triglyceride accumulation, and adipogenesis-specific protein and transcription factor expression can serve as effective anti-obesity therapeutics.

Seaweeds are multicellular marine protists that can be observed with the naked eye. They are classified into green, brown, and red algae according to their photosynthetic pigment composition [21,22]. Recently, many studies have explored the nutritional benefits of seaweeds. Seaweeds contain abundant nutrients, such as polyphenols, vegetable proteins, vitamins, and minerals. Thus, the development of various functional materials using seaweed extracts has garnered much attention [23,24,25]. Recently, polysaccharides extracted from seaweeds were reported to exhibit antioxidant, anti-inflammatory, anticancer, and immune activities; thus, seaweeds have attracted much research interest as a functional biomaterial and food [26,27].

Au nanostructures (AuNSs) are used in numerous pharmaceutical fields because of their uniform particle size distribution, easy surface modification, and excellent stability [28,29]. Recently, research on AuNP synthesis with seaweed extract without the use of organic solvents has been actively conducted. Seaweed extracts can be used as a reducing agent and stabilizer and offer the advantage of bioactivity when applied to the human body [30,31,32]. *Plocamium telfairiae* (PT) is a red alga inhabiting coastal waters. It contains phycoerythrin, is red, and is a macroalga that evolves into a complex multicellular system, such as agar. The antioxidant activity of polyphenols in PT has been documented, and the algal extract has also been reported to produce anticancer and anti-obesity effects [33,34,35].

To this end, in the present study, we investigated whether PT-functionalized AuNSs (PT-AuNSs) can alleviate MDI-induced mature adipocyte activation and whether the mechanism underlying their possible anti-adipogenic effects is through the inhibition of phospholipase D1 (PLD1)-mediated adipogenesis.

## 2. Results and Discussion

### 2.1. Physicochemical Characterization of PT-AuNSs

General component analysis revealed that the PT extract is composed of sterols (0.29 ± 0.04%), lipids (10.12 ± 0.07%), carbohydrates (18.65 ± 0.52%), proteins (25.69 ± 0.09%), moisture (5.32 ± 0.14%), ash (25.97 ± 0.24%), and polyphenols (5.49 ± 0.07%). The major biomolecules of the PT extract are sterols, such as sulfated polysaccharides, fucosterol, and cholesterol; as well as surface-bound proteins and nanoparticles, such as muramic acid, glucuronic acid, alginic acid, and vinyl derivatives. These can act as reducing and capping agents as well as supply residual amino acids. Furthermore, they help reduce, encapsulate, and stabilize the PT-AuNS. The maximum local surface plasmon resonance peak of PT-AuNS was confirmed at 526 nm via ultraviolet–visible (UV–Vis) spectroscopy (data not shown). The size, zeta potential, and polydispersity index (PDI) of PT-AuNS were analyzed using dynamic light scattering (DLS). The average size of PT-AuNS was 15.36 ± 0.10 nm, and its zeta potential was −35.85 ± 1.36 mV, confirming the high stability of the structure. Finally, to confirm the nano-delivery and colloidal characteristics of PT-AuNS, the PDI was determined. A PDI of <0.7 indicates very high stability in the nano-delivery/colloidal system. The PDI of PT-AuNS was 0.355. Next, the shape, size, dispersion, and crystalline structure of PT-AuNS were analyzed using high-resolution transmission electron microscopy (HR-TEM), high-angle annular dark-field (HAADF) imaging, selected area electron diffraction (SAED), and energy-dispersive X-ray spectroscopy (EDX). In HR-TEM images, in addition to pentagonal and hexagonal shapes, most of the nanostructures were spherical and well dispersed without lumps. The diameter of PT-AuNS was approximately 28.32–32.69 nm (Figure 1A,B). The distribution of Au in PT-AuNS in the HAADF mode was verified using red gold particle images (Figure 1C,D). In the SAED mode, the bright rings correspond to the (111), (200), (220), and (311) lattice planes. The structure of PT-AuNS was verified to be a face-centered cube (Figure 1E,F). In the EDX spectrum of PT-AuNS, gold peaks of 0.2–0.5, 2–2.4, and 9.5–9.8 keV were identified (Figure 1G). In addition, Fourier-transform infrared (FTIR) spectroscopy was used to explore the potential biomolecules of PT extracts as reducing, capping, and stabilizing agents. The peaks at 3430 and 1613 cm^−1^ for PT and those at 3435 and 1615 cm^−1^ PT-AuNS correspond to the NH stretching vibration of the amine group. The peak at 1384 cm^−1^ for PT and that at 1400 cm^−1^ for PT-AuNS correspond to the stretching of the –OH bond in the polyphenol or alcohol group. The peaks at 600 and 893 cm^−1^ for PT and those at 602 and 894 cm^−1^ for PT-AuNS correspond to the aliphatic C-H stretching vibration of the alkyne group (Figure 1H,I). These results indicate that the potential biomolecules and functional chains in PT extracts—including amino acids, primary and tertiary amines, polyphenols, and polysaccharides, among others—act as capping, stabilizing, and reducing agents during PT-AuNS synthesis.

### 2.2. Effects of PT-AuNS on Oil Red O Staining Activity and TG Accumulation

Cell viability assays confirmed that PT extract and PT-AuNS did not produce any effect on pre- and mature adipocytes compared with the respective controls. PT extract was not toxic to pre- and mature adipocytes at concentrations of up to 200 μg·mL^−1^. Similarly, PT-AuNS did not affect the viability of pre- and mature adipocytes at concentrations up to 200 μg·mL^−1^ (Figure 2A,B). Based on these results, the PT extract and PT-AuNS were used at concentrations ranging from 10 to 200 μg·mL^−1^ in subsequent experiments. Oil Red O staining analysis was performed to investigate whether the PT extract and PT-AuNS affected lipid droplet accumulation in pre- and mature adipocytes. To set the initial treatment concentration, pre- and mature adipocytes were pretreated with 10, 20, 40, 80, 150, or 200 μg·mL^−1^ of the PT extract and PT-AuNS and then subjected to Oil Red O staining. PT extract pretreatment reduced Oil Red O staining activity in mature adipocytes in a dose-dependent manner, with the maximum reduction of 58.3% at 200 μg·mL^−1^. Moreover, PT-AuNS pretreatment reduced Oil Red-O staining activity by 82.8% at 20 μg·mL^−1^, and at higher concentrations, the Oil Red-O staining activity of mature and preadipocytes was comparable (data not shown). Based on these results, 200 μg·mL^−1^ PT extract and 20 μg·mL^−1^ PT-AuNS were selected for the comparative study of the effect of PT extract and PT-AuNS on adipogenesis. In preadipocytes, the cytoplasm maintains the same form as that in fibroblasts, and the following differentiation into mature adipocytes, spherical, and round lipid droplets are formed on the cell membrane [36]. We cultured pre- and mature adipocytes for 9 days in the presence of PT extract and PT-AuNS and then analyzed cell morphology and lipid droplets using bright-field imaging. The change from mature adipocytes to globular cells and formation of lipid droplets induced by MDI increased in controls. However, in cells pretreated with PT extract and PT-AuNS, the change from mature adipocytes to globular cells and the formation of lipid droplets were inhibited (Figure 2C). MDI-treated preadipocytes started to differentiate into mature adipocytes, and Oil Red O staining showed high intracellular lipid droplet accumulation on day 9; however, these effects were reduced following pretreatment with PT extract and PT-AuNS. Interestingly, the decrease in Oil Red O staining activity was greater after treatment with 20 μg·mL^−1^ PT-AuNS than the 200 μg·mL^−1^ PT extract, but was not affected by conventionally synthesized citrate-AuNS (Figure 2D,E). Triglycerides stored in adipose tissue are the markers of lipid accumulation and most adipocytes store triglycerides in lipid droplets. Thus, the effect of PT extract and PT-AuNS on triglyceride content was assessed in pre- and mature adipocytes. Both PT extract (200 μg·mL^−1^) and PT-AuNS (20 μg·mL^−1^) decreased MDI-induced triglyceride content by approximately 46.8% and 78.6%, respectively (Figure 2F). Collectively, these results indicate that PT extract and PT-AuNS inhibit the production of lipid droplets and triglycerides, which are major markers involved in adipogenesis, and that PT-AuNS exhibits a stronger activity than PT extract.

### 2.3. Effect of PT Extract and PT-AuNS on Nile Red Staining Activity

Lipid droplets act as storehouses for excess neutral lipids, typically triacylglycerols or cholesteryl esters. Abnormal accumulation of lipid droplets was observed in mature adipocytes [37]. Nile red (excitation/emission: 550/640 nm) exhibits intense fluorescence and serves as a sensitive stain for the detection of cytoplasmic lipid droplets. Flow cytometry and confocal microscopy are powerful tools for analyzing lipid droplets through Nile red staining. Figure 3A shows a panel of images showing lipid droplets in pre- and mature adipocytes incubated with Nile Red for 1 h. In mature adipocytes, the number and size of lipid droplets were significantly higher than those in preadipocytes. However, pretreatment with the PT extract and PT-AuNS significantly reduced the number and size of these lipid droplets. In preadipocytes treated with PT extract and PT-AuNS, there were no changes in the number and size of lipid droplets compared with control values. To confirm these visual findings, we quantified cellular lipid droplet levels in pre- and mature adipocytes using flow cytometry. As shown in Figure 3B,C, the fluorescence intensity of cells treated with PT extract (200 μg·mL^−1^) and PT-AuNS (20 μg·mL^−1^) decreased by 56.5% and 84.8%, respectively. Furthermore, Nile-red activity was not affected by conventionally synthesized citrate-AuNS (20 μg·mL^−1^) treatment. Thus, the results of flow cytometry were consistent with those of Oil Red O staining.

### 2.4. Effects of PT Extract and PT-AuNS on the Transcript and Protein Expression of Adipogenic Marker

The expression and activation of many transcription factors and adipocyte-specific genes are involved in adipocyte differentiation and lipid deposition [8]. Following MDI induction of preadipocytes, which no longer grow owing to the formation of intercellular hyperjunctions, differentiation into mature adipocytes starts. During adipocyte differentiation and lipid deposition, SREBP-1c, PPAR-γ, and C/EBPα are the major adipogenic transcription factors, and these genes regulate the transcription of FAS and aP2, which are responsible for morphological changes and lipid accumulation in adipocytes [12,18]. SREBP-1c augments the expression of genes involved in adipogenesis, particularly fatty acid biosynthesis and triglyceride maturation. PPAR-γ induces insulin sensitization and promotes glucose metabolism, thereby promoting C/EBPα expression and activation and ultimately inducing Fas and aP2 to promote adipocyte differentiation and lipid deposition. FAS is a multienzyme protein catalyzing fatty acid synthesis, while aP2 promotes the transport of lipids to specific cellular compartments through the storage of lipid droplets [19,36]. To confirm the direct effects of PT extract and PT-AuNS on lipid accumulation, we investigated their effects on the expression of C/EBPα, PPARγ, SREBP-1, FAS, and aP2. Compared with that in preadipocytes, the protein expressions of C/EBPα, PPARγ, SREBP-1, FAS, and aP2 were significantly upregulated in MDI-induced mature adipocytes. However, their expression was significantly downregulated following pretreatment with PT extract (200 μg·mL^−1^) and PT-AuNS (20 μg·mL^−1^) (Figure 4A). We further performed real-time PCR to determine whether the suppression of C/EBPα, PPAR, SREBP-1, FAS, and aP2 protein levels by PT extracts and PT-AuNS was mediated through transcriptional regulation. Experimental data showed that PT extract (200 μg·mL^−1^) and PT-AuNS (20 μg·mL^−1^) reduced the transcript levels of C/EBPα, PPARg, SREBP-1, FAS, and aP2 in MDI-induced mature adipocytes (Figure 4B). In addition, the transcript and protein expression of C/EBPα, PPARg, SREBP-1, FAS, and aP2 in preadipocytes was not significantly different from that in controls following pretreatment with PT extract (200 μg·mL^−1^) or PT-AuNS (20 μg·mL^−1^). These results confirm that PT extract and PT-AuNS exert anti-adipogenic effects by suppressing the transcription and translation of adipogenic factors.

### 2.5. Effects of PT Extract and PT-AuNS on the Mitochondrial Thermogenesis in Mature Adipocyte

In mammals, adipose tissue contains brown adipocytes, which consume energy to generate heat; as well as white adipocytes, which store energy for fat accumulation [38]. A characteristic of brown adipocytes is that they possess a large number of mitochondria, and protons in the inner mitochondrial membrane diffuse through glucose consumption and fat oxidation to generate heat. Uncoupling protein 1 (UCP1), PR domain containing 16 (PRDM16), and peroxisome proliferator-activated receptor-gamma coactivator 1α (PGC1α) play important roles in mitochondrial biosynthesis and energy expenditure during thermogenesis in brown adipocytes [39,40]. Specifically, UCP1 plays an important role in energy expenditure by adipocytes by supporting mitochondrial biosynthesis and thermogenesis in the mitochondrial inner membrane via direct binding with free fatty acids. Meanwhile, PGC-1a is an electron coactivator regulating transcription factors involved in mitochondrial respiration and biosynthesis and stimulates tricarboxylic acid cycle expression, peroxisomal activity, and mitochondrial fatty acid oxidation. Finally, PRDM16 enhances the expression of genes involved in mitochondrial biosynthesis through interaction with PGC-1a and promotes brown adipogenesis in adipocytes [41,42,43]. Here, immunofluorescence screening was performed to confirm mitochondrial biogenesis and UCP-1 expression, which are characteristics of brown adipogenesis. Mature adipocytes were stained with MitoTracker (red), which specifically binds mitochondria, followed by FITC (green)-conjugated UCP1 antibody. Following pretreatment with PT extract (200 μg·mL^−1^) and PT-AuNS (20 μg·mL^−1^), more intense red fluorescence was observed in the cytoplasm of mature adipocytes, and UCP1 was colocalized in the mitochondria and cytoplasm (Figure 5A). To further investigate the mechanism of the browning effect of PT extract and PT-AuNS, we measured the transcript and protein expression of UCP1, PRDM16, and PGC1a. The protein levels of UCP1, PRDM16, and PGC1a were evaluated in mature adipocytes. Furthermore, pretreatment of mature adipocytes with PT extract (200 μg·mL^−1^) and PT-AuNS (20 μg·mL^−1^) increased the expression of UCP1, PRDM16, and PGC1a. Likewise, pretreatment with PT extract (200 μg·mL^−1^) and PT-AuNS (20 μg·mL^−1^) significantly upregulated the transcript expression of UCP1, PRDM16, and PGC1a (Figure 5B). These results suggest that PT extract and PT-AuNS upregulated mitochondrial biogenesis in mature adipocytes and promoted the expression of brown adipogenesis marker genes.

### 2.6. Anti-Adipogenesis Effects of PT Extract and PT-AuNS via PLD1 Regulation

Adipocyte differentiation is regulated through the mammalian target of rapamycin (mTOR) signaling cascade. Phospholipase D1 (PLD1) promotes cellular differentiation by activating the mTOR signaling pathway. PLD1 is a hydrolase that catalyzes the conversion of phosphatidylcholine to phosphatidic acid (PA) and choline. PLD1 and PA are involved in the differentiation and development of various cells through the activation of the mTOR signaling pathway. PLD1 specifically activates the mTOR signaling pathway by stimulating mitogens or amino acids [44,45]. Moreover, it regulates cellular differentiation processes, such as myogenesis and decidualization. PLD1 expression in adipocytes is correlated with PPARγ and C/EBPα expression [46]. To determine whether PLD1 knockdown is required for induction of adipogenic differentiation, we transfected pre- and mature adipocytes with PLD1-specific siRNA and negative control siRNA. Interestingly, adipogenic differentiation of PLD1-specific siRNA-transfected mature adipocytes was significantly increased by mRNA levels of C/EBPα, PPARγ, and FAS (Figure 6A). To determine whether PLD1 could mediate the PT extract- and PT-AuNS-induced anti-adipogenesis, adipocytes were transfected with mouse PLD1-specific siRNA or negative control siRNA and then induced with MDI differentiation medium. As shown in Figure 6B, PLD1 transcript expression was knocked down through PLD1 siRNA transfection. We investigated whether PLD1 deficiency upregulated the expression of C/EBPα, PPARγ, and FAS. Indeed, the transcript and protein expression levels of C/EBPα, PPARγ, and FAS were increased in the PLD1 siRNA-transfected group, and these effects were inhibited by PT extract and PT-AuNS (Figure 6C–E). In addition, Nile red staining was performed to investigate whether PLD1 expression is required for lipid droplet formation. Lipid droplet formation increased due to the lack of PLD1, and the size of droplets formed was also increased. These observations were further confirmed by the suppressive effects of PT extract and PT-AuNS (Figure 6F,G). Therefore, in mature adipocytes deficient in PLD1, differentiation was more promoted through FAS upregulation through PPAR-γ and C/EBPα induction than in mature adipocytes, and it was confirmed that this effect was indirectly inhibited by PT-AuNS. These results suggest that PT-AuNS can mitigate or rescue PLD1-knockdown related increases in adiposity.

## 3. Materials and Methods

### 3.1. Reagents

The following reagents were obtained from Sigma-Aldrich (Merck KGaA, Darmstadt, Germany): HAuCl_4_·3H_2_O, ethanol, Cell Counting Kit-8 (CCK-8), protease inhibitor, DAPI mounting media, and X-treme GENE siRNA transfection reagent. Mouse 3T3-L1 preadipocytes (CL-173, Lot:70047755) were obtained from the American Type Culture Collection (ATCC, Manassas, VA, USA). The 3T3-L1 differentiation kit and lipid (Oil Red O) staining kit were obtained from BioVision (Milpitas, CA, USA). The Nile Red staining kit and triglyceride colorimetric assay kit were obtained from Abcam (Cambridge, MA, USA). The following reagents were obtained from Thermo Fisher Scientific Life Sciences (Waltham, MA, USA): Dulbecco’s modified Eagle’s medium/nutrient mixture F-12 (DMEM/F12), fetal bovine serum (FBS), phosphate-buffered saline (PBS), penicillin and streptomycin, eight-well chamber slides, M-PER™ Mammalian Protein Extraction Reagent, Pierce ECL Western blotting substrate, PureLink RNA Mini Kit, high-dose cDNA reverse kit, SYBR Green qPCR Master Mix, PLD1-specific siRNA, and negative control siRNA.

### 3.2. Preparation of PT Extract

PT was acquired off the coast of Jeju Island. The collected samples were sterilized and dried in a cool place. PT was purchased from Jeju Technopark Biodiversity Research Institute (gift certificate sample number: JBRI-16041). PT specimens were stored in the Herbarium of the Jeju Institute of Biological Diversity, and the identification of deposited PTs was performed by Dr. Wookjae Lee (Jeju Technopark, Jeju, Korea) [33]. The samples were dehydrated and ground to a fine powder using an electric mixer (HMF-3100S, Hanil Electric, Seoul, Korea). A 40–50 mesh standard test sieve was used to obtain the smallest possible size of the PT powder. To prepare the PT extract, 10 g of well-ground PT powder and 300 mL of 80% ethanol were mixed, extracted at room temperature for 12 h, and centrifuged at 6000× *g* for 10 min to obtain the supernatant. The supernatant was filtered and concentrated using a rotary vacuum evaporator (Buchi Rotavapor R-144, Buchi Labortechnik, Flawil, Switzerland). The obtained PT extract was freeze-dried, and the PT powder was preserved at −80 °C for subsequent experiments. The PT powder was dissolved in an aqueous solution at a concentration of 4 mg·mL^−1^ to use in the PT-AuNS medium. The solution was filter sterilized using a 0.2 μm syringe filter.

### 3.3. PT-AuNS Synthesis

AuNSs were synthesized by optimizing the concentration, temperature, and reaction time of the PT extract and metal precursor. First, aqueous HAuCl_4_·3H_2_O solution (1M) was added to the PT extract (4 mg·mL^−1^), and the mixture was incubated in a water bath at 80 °C for 15 min [47]. The tube containing PT-AuNSs was placed on ice for 5 min. When the color of the suspension changed to red-violet, PT-AuNSs were successfully synthesized.

### 3.4. Characterization of PT-AuNS

The formation of PT-AuNS was examined using the Ultrospec 6300 pro UV–Vis spectrophotometer (Amersham Biosciences, Buckinghamshire, UK) in the wavelength range of 300–800 nm. The size, zeta potential, and PDI of PT-AuNS were determined using the Zetasizer Nano-ZS90 sample delivery system (Malvern Panalytical, Malvern, UK). The morphology and particle size distribution of PT-AuNS were determined using HR-TEM TALOS F200X (Thermo Scientific, Waltham, MA, USA) at an electron potential of 200 kV. To confirm that PT-AuNS were synthesized via PT extraction, FTIR spectra were obtained using the Spectrum GX spectrometer (Perkin Elmer Inc., Boston, MA, USA) prepared with potassium bromide pellets [47].

### 3.5. Pre- and Mature Adipocyte Culture and Treatment

Mouse 3T3-L1 preadipocytes were cultured in DMEM/F12 containing 10% FBS and 1% penicillin–streptomycin. To obtain mature adipocytes, preadipocytes were incubated as described above until reaching confluence. After replacing the medium after 2 days, differentiation was induced using the 3T3-L1 differentiation kit. The cells were incubated for 3 days in the provided differentiation cocktail (MDI differentiation medium: 1 μM dexamethasone, 500 μM IBMX, 1.5 μg·mL^−1^ insulin, and 1 μM rosiglitazone) [10]. The adipocytes cultured for 3 days were cultured for an additional 2 days in DMEM/F12 containing 1.5 μg·mL^−1^ insulin. Thereafter, the culture medium was replaced once every 2 days. To evaluate its anti-adipogenic effects, PT-AuNS (10–20 μg·mL^−1^) was administered 2 h before exposure to MDI differentiation medium for 9 days (Figure 7).

### 3.6. Viability of Pre- and Mature-Adipocytes

CCK-8 was used for the cell viability assays of pre- or mature adipocytes. The cells were seeded in a 48-well plate at a density of 1×10^4^ cells·L^−1^ and incubated at 37 °C under 5% CO_2_. After 2 days, the cells were pretreated with PT-AuNS at 0, 1, 2.5, 5, 10, 20, and 40 μg·mL^−1^ for 2 h and then either treated or not with an MDI differentiation medium. The CCK-8 reagent was added to each well at a concentration of 1/100, and the plates were wrapped in foil and incubated for 4 h. For quantification, absorbance was measured at 450 nm using the Wallac VICTOR plate reader (Perkin Elmer Corp., Nerwalk, CT, USA) [47].

### 3.7. Lipid Droplet Measurement Using Oil Red O and Nile Red Staining

After inducing adipocyte differentiation for 9 days, the cells were stained on day 9 with a lipid (Oil Red O) staining kit and Nile red staining kit. Briefly, mature adipocytes were washed two times with PBS, fixed with 10% formalin, and stained with Oil Red O solution for 1 h at room temperature. After 1 h, the excess staining solution was removed, and images of stained lipid droplets were obtained with an optical microscope. For quantification of lipid droplets, the Oil Red O staining reagent remaining in the cells was dissolved in isopropanol and separated, and absorbance was measured at 490 nm. For Nile red staining, 4000 preadipocytes were cultured on eight-well chamber slides, and then mature adipocytes were induced using the method described above. Mature adipocytes were rinsed with PBS, fixed with 10% formalin buffer, and stained with Nile red solution for 0.5 h at room temperature. Images of stained lipid droplets were captured with a confocal microscope (LSM 800, Carl Zeiss, Jena, Germany), and the droplets were quantified based on the fluorescence intensity of a fluorescent dye with a flow cytometer (Fit NxT Flow Cytometer; Thermo Fisher Scientific, Inc. Waltham, MA, USA) [43].

### 3.8. Triglyceride Analysis

To control intracellular triglyceride levels, preadipocytes were inoculated at a density of 2 × 10^6^ cells per well in six-well plates. Reagent treatment and differentiation induction were performed according to the method described above. A triglyceride colorimetric assay kit was used to measure triglyceride content. The protein concentration of the cell lysate was determined using a Bio-Rad protein assay kit [43].

### 3.9. Reverse Transcription-Quantitative Polymerase Chain Reaction (RT-qPCR)

Total RNA was extracted from adipocytes using the PureLink RNA Mini Kit, and RNA concentration was determined using a NanoDrop spectrophotometer (Thermo Scientific, Waltham, MA, USA). Subsequently, cDNA was synthesized using a high-dose cDNA reverse kit. Relative quantification of gene expression was performed using SYBR Green qPCR Master Mix and Bio-Rad Chromo4 (Applied Biosystems, Hercules, CA, USA). Samples were amplified in triplicate in 96-well plates, with 40 cycles at 95 °C for 15 s and 58 °C for 40 s. The primers used are listed in Table 1. Data were analyzed using the 2^−^^ΔΔCT^ method [47].

### 3.10. Western Blotting

Adipocytes were lysed using a M-PER™ Mammalian Protein Extraction Reagent containing a protease inhibitor. The mixture was placed on ice for 30 min, shaken every few minutes, and centrifuged at 12,000 rpm and 4 °C for 10 min. The supernatant was collected, and protein concentration was measured using a Bio-Rad protein assay kit (Bio-Rad Laboratories, Inc., Hercules, CA, USA). Mini-PROTEAN Precast Gels (Bio-Rad Laboratories, Inc.) were prepared, and the samples (30 μg) were loaded following protein denaturation. A Hybond polyvinylidene difluoride membrane (Cytiva, Amersham, England) was selected, and the proteins were transferred onto the membrane. The membranes were blocked for 1 h and incubated overnight at 4 °C with diluted primary antibodies. The primary antibodies used are listed in Table 2. The membranes were then washed three times and incubated with horseradish peroxidase-conjugated secondary antibody for 1 h at room temperature. After washing, the membranes were incubated with enhanced Pierce ECL Western blotting substrate. The relative intensities of bands were determined using ImageQuant TL (version 8.1; Cytiva, Amersham, UK).

### 3.11. Immunofluorescence Assay

For immunofluorescence assays, 4000 preadipocytes were cultured on eight-well chamber slides, and then mature adipocytes were induced using the method described above. The adipocytes were treated with 0.1% Triton X-100 serum for 30 min and then fixed with 4% paraformaldehyde for 30 min. Permeabilized adipocytes were blocked with 1% BSA in PBST for 1 h and incubated overnight with antibodies against UCP1 (Alexa Fluor 488 Conjugate, 1:200, Cell Signaling Technology, Beverly, MA, USA) at 4 °C. The adipocytes were washed three times with PBS and then incubated with DAPI mounting media for 0.5 min at 37 °C in the dark. The adipocytes were visualized using the Zeiss LSM 800 confocal laser scanning microscope (Zeiss, Oberkochen, Germany).

### 3.12. PLD1 Knockdown Using siRNA Transfection

Adipocytes were transfected with mouse PLD1-specific siRNA and negative control siRNA using X-treme GENE siRNA transfection reagent according to the manufacturer’s instructions. The X-treme GENE siRNA transfection reagent, control siRNA, and si-PLD1 were mixed with Opti-MEM I Reduced Serum Medium and then activated for 20 min at room temperature, transfection mixture with control siRNA, and si-PLD1 were then added to the adipocytes. After 24 h of incubation, fresh medium was added and the adipocytes were grown for additional 48 h. Gene silencing was confirmed using RT-qPCR. To confirm the anti-adipogenic effect, PT-AuNS (10–20 μg·mL^−1^) was administered for 9 days and 2 h before exposure to MDI differentiation medium.

### 3.13. Statistical Analysis

All analyses were conducted using SPASS 22.0 software (SPSS Inc., New York, NY, USA), and GraphPad Prism 4.0 0 (GraphPad, San Diego, CA, USA) was used for graphing. Quantitative data were expressed as mean ± standard deviation (mean ± SD), and *p* < 0.05 was statistically significant. Cell viability, oil-red, Nile-red, triglyceride rate of adipocyte, the relative expression of C/EBPα, PPARγ, SREBP-1, FAS, aP2, UCP1, PRDM16, and PGC1α mRNA and proteins among different groups were analyzed using one-way analysis of variance (ANOVA) followed by Tukey’s post-hoc test.

## 4. Conclusions

Nanostructures are among the most advanced research areas and have found wide applications owing to their unique properties. Metal and metal oxide nanostructures synthesized using natural resources possess excellent physical, chemical, electrical, optical, and biofunctional characteristics. The novelty of the present study lies in that our proposed protocol of nanostructure fabrication is economical and rapid relative to the current methodologies that require harmful chemicals and high energy. In the present work, we synthesized novel PT-AuNSs from the seaweed *P. telfairiae* in an environmentally friendly manner while avoiding the use of toxic compounds. The produced PT-AuNSs were characterized using a series of analytical techniques—including UV–Vis spectroscopy, DLS, FTIR spectroscopy, HR-TEM, and EDS—to determine their physical and chemical properties as well as their structure and size. Furthermore, we performed evaluations at the cellular level using pre- and mature adipocytes, focusing on the mechanism of adipogenesis. The effects of PT-AuNSs on pre- and mature adipocytes have not been reported, and there have been no comparative studies of PT extracts and PT-AuNSs. Here, we demonstrated the anti-adipogenic effects of PT extract and PT-AuNS on mature adipocytes. Interestingly, PT-AuNSs (20 μg·mL^−1^) exhibited a more potent anti-adipogenic activity than PT extract (200 μg·mL^−1^). Finally, the anti-adipogenic activity of PT extracts and PT-AuNS in mature adipocytes was mediated through PLD1.

## Figures and Tables

**Figure 1 marinedrugs-20-00421-f001:**
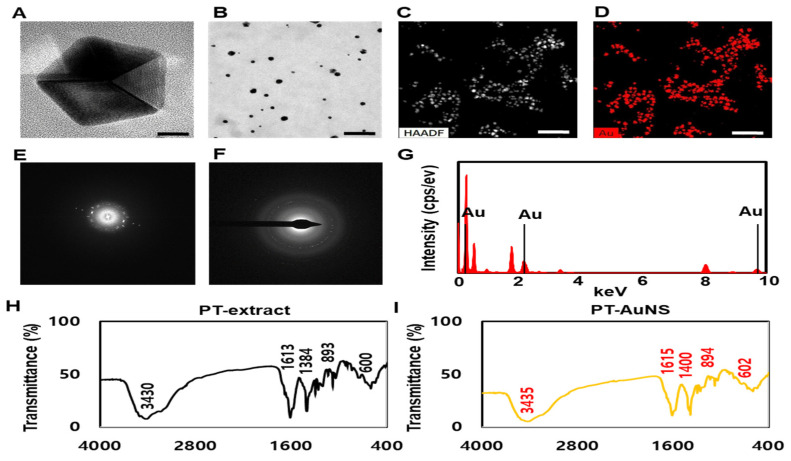
HR-TEM and FTIR spectroscopy of PT-AuNS. HR-TEM images at low (**A**, Scale bar = 10 nm.) and high (**B**, Scale bar = 200 nm.) magnification. HAADF (**C**,**D**, Scale bar = 200 nm.) and SAED (**E**,**F**) images of PT-AuNS. EDS elemental map (**G**) of PT-AuNS. FTIR spectra of PT extract (**H**) and PT-AuNS (**I**).

**Figure 2 marinedrugs-20-00421-f002:**
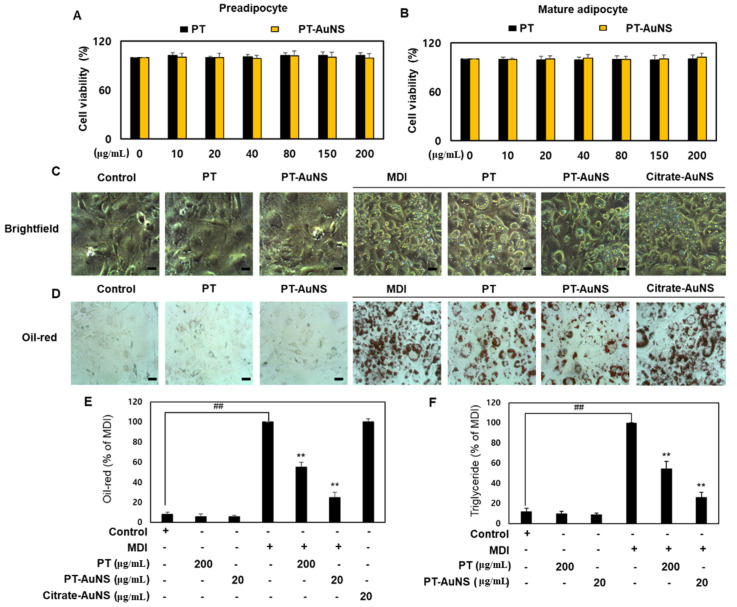
Morphological changes and inhibitory effects of PT extract, PT-AuNS, or citrate-AuNS in pre- and mature adipocytes. Effects of PT extract and PT-AuNS on the viability of preadipocytes (**A**) and mature adipocytes (**B**). (**C**) Morphological changes. Scale bar = 100 μm. (**D**) Oil Red O staining activity. Scale bar = 20 μm. (**E**) Quantitative Oil Red O staining activity of data presented in (**D**). (**F**) Triglyceride content. The experiment was repeated three times, and the data were shown as mean ± standard deviation. (*n* = 3 for every group, ** *p* < 0.01 vs. MDI-differentiated group; ^##^
*p* < 0.01 vs. control).

**Figure 3 marinedrugs-20-00421-f003:**
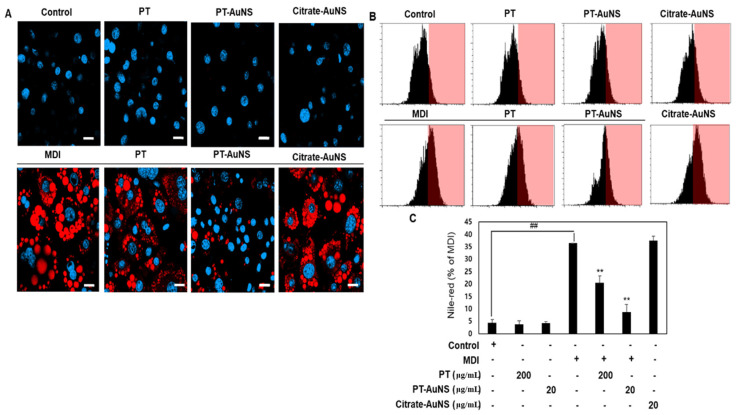
Nile red staining activity of cells treated with PT extract, PT-AuNS, or citrate-AuNS. (**A**) Nile red staining activity using confocal microscopy. Scale bar = 20 μm. (**B**) Nile-red staining activity using flow cytometry. (**C**) Quantitative analysis of Nile red staining activity data presented in (**B**). The experiment was repeated three times, and the data were shown as mean ± standard deviation. (*n* = 3 for every group, ** *p* < 0.01 vs. MDI-differentiated group; ^##^
*p* < 0.01 vs. control).

**Figure 4 marinedrugs-20-00421-f004:**
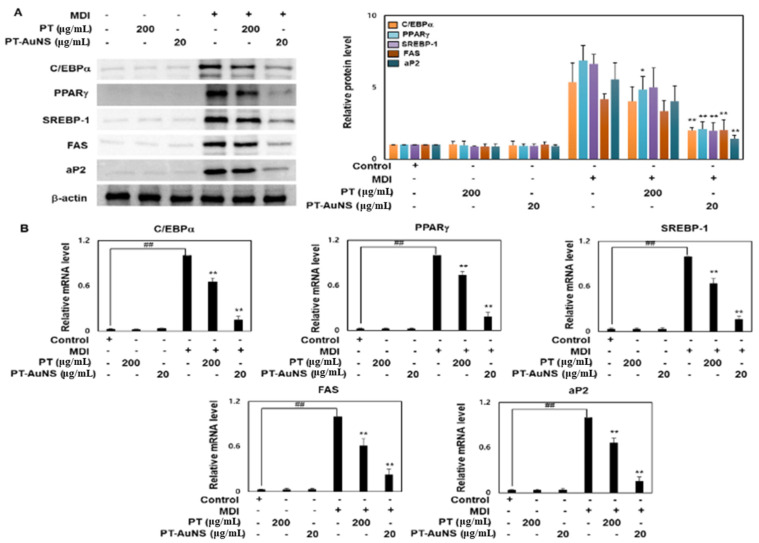
Downregulation of C/EBPα, PPARγ, SREBP-1, FAS, and aP2 following pretreatment with PT extract and PT-AuNS. (**A**) Expression levels of adipogenesis-related proteins, as measured using Western blotting. (**B**) Transcript expression levels of adipogenesis-related genes, as measured using RT-qPCR. The experiment was repeated three times, and the data were shown as mean ± standard deviation. (*n* = 3 for every group, ^*^
*p* < 0.05 and ** *p* < 0.01 vs. MDI-differentiated group; ^##^
*p* < 0.01 vs. control).

**Figure 5 marinedrugs-20-00421-f005:**
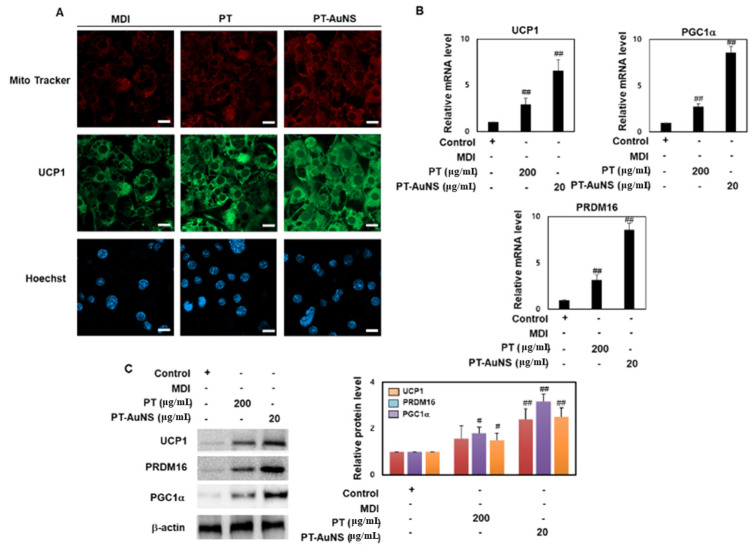
Upregulation of mitochondrial thermogenesis following pretreatment with PT extract and PT-AuNS. (**A**) Effects of PT extract and PT-AuNS on the expression of MitoTracker (red) and anti-Ucp1 antibody (green) in MDI-induced adipocytes. Scale bar = 20 μm. (**B**) Transcript expression levels of brown adipogenesis marker genes, as measured using RT-qPCR. (**C**) Protein expression levels of brown adipogenesis marker genes, as measured using Western blotting. The experiment was repeated three times, and the data were shown as mean ± standard deviation. (*n* = 3 for every group, ^#^
*p* < 0.05 and ^##^
*p* < 0.01 vs. control).

**Figure 6 marinedrugs-20-00421-f006:**
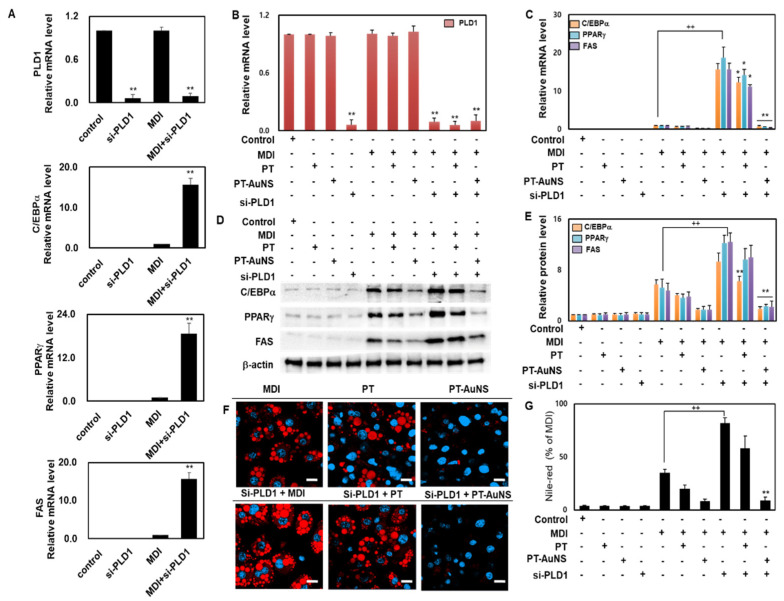
Anti-adipogenic effects of PT extract and PT-AuNS through PLD1. (**A**) RT-qPCR was used to detect the transcript expression of PLD1, C/EBPα, PPARγ, and FAS (*n* = 3 for every group, ** *p* < 0.01 vs. MDI-differentiated group). (**B**) RT-qPCR was used to detect the transcript expression of PLD1 (*n* = 3 for every group, ** *p* < 0.01 vs. control). (**C**) Transcript expression levels of adipogenesis-related genes, as measured using RT-qPCR (*n* = 3 for every group, ^++^ *p* < 0.01 vs. MDI-differentiated group; * *p* < 0.05 and ** *p* < 0.01 vs. MDI-differentiated group). (**D**) Expression levels of adipogenesis-related proteins, as measured using Western blotting. (**E**) Quantitative results of adipogenesis-related protein expression levels (*n* = 3 for every group, ^++^ *p* < 0.01 vs. MDI-differentiated group; ** *p* < 0.01 vs. MDI-differentiated group). (**F**) Nile red staining activity using confocal microscopy. Scale bar = 20 μm. (**G**) Quantitative Nile red staining activity using flow cytometry (*n* = 3 for every group, ^++^
*p* < 0.01 vs. MDI-differentiated group; ** *p* < 0.01 vs. MDI-differentiated group). The experiment was repeated three times, and the data were shown as mean ± standard deviation.

**Figure 7 marinedrugs-20-00421-f007:**
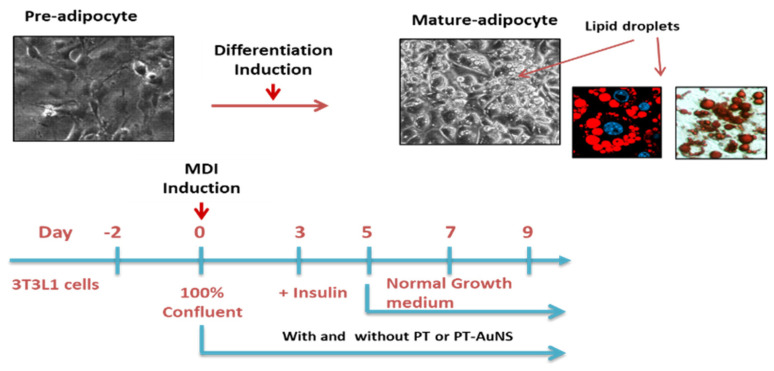
Experimental design of pre- and mature adipocyte culture and PT-AuNS treatment.

**Table 1 marinedrugs-20-00421-t001:** Primer sequences used for the qPCR analysis.

Gene	Forward Primer (5′→3′)	Reverse Primer (3′→5′)
** *C/EBP* ** **α**	TGGTGATTTGTCCGTTGTCT	GGA AACCTGGCCTGTTGTAAG
** *PPAR* ** **γ**	GGTGATTTGTCCGTTGTCT	GCTTCAATCGGATGGTTCTTC
** *SREBP-1* **	TAGAGCATATCCCCCAGGTG	GGTACGGGCCACAAGAAGTA
** *FAS* **	GCTGCGGAAACTTCAGGAAAT	AGAGACGTGTCACTCCTGGACTT
** *aP2* **	GGATTTGGTCACCATCCGGT	TTCACCTTCCTGTCGTCTGC
** *UCP1* **	CCTGCCTCTCTCGGAAACAA	GTAGCGGGGTTTGATCCCAT
** *PRDM16* **	CAGCACGGTGAAGCCATTC	GCGTGCATCCGCTTGTG
** *PGC1α* **	ATGTGCAGCCAAGACTCTGTA	CGCTACACCACTTCAATCCAC
** *GAPDH* **	AGGTCGGTGTGAACGGATTTG	TGTAGACCATGTAGTTGAGGTCA

**Table 2 marinedrugs-20-00421-t002:** Antibodies used for Western blotting.

Antibody	Company	Catalog	Species	Dilution
** *C/EBPα* **	Cell Signaling	#8178	Rabbit	1:500
** *PPARγ* **	Cell Signaling	#2435	Rabbit	1:500
** *SREBP-1* **	Santa Cruz	sc-365513	Mouse	1:1000
** *FAS* **	Cell Signaling	#3180	Rabbit	1:500
** *aP2* **	Cell Signaling	#2120	Rabbit	1:500
** *UCP1* **	Santa Cruz	sc-293418	Mouse	1:1000
** *PRDM16* **	Abcam	ab106410	Mouse	1:500
** *PGC1α* **	Santa Cruz	sc-518025	Mouse	1:500
** *β-actin* **	Santa Cruz	sc-47778	Goat	1:1000

## Data Availability

The data that support the findings of this study are available from the corresponding author upon reasonable request.

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
