# Peer review of "Characterization of Plocamium telfairiae Extract-Functionalized Au Nanostructures and Their Anti-Adipogenic Activity through PLD1"

_marinedrugs, 2022, doi:10.3390/md20070421_

Round 1

Reviewer 1 Report

In their manuscript “Characterization of Plocamium telfairiae Extract-Functionalized Au Nanostructures and their Anti-Adipogenic activity through PLD1” Park et al. describe a method for functionalizing Au nanostructures with ethanolic extract from seaweed, and testing this for its anti-adipogenic effects. While this functionalization method is of interest to the field, the author’s conclusion that the anti-adipogenic effects are through PLD1 are not supported by the data presented. The authors show that the PT and PT-AuNS treatments are able to reduce many markers of adiposity in MDI-treated fibroblasts, including Oil Red O and Nile Red staining as well as C/EBPα, PPARγ, SREBP-1, FAS, and aP2 mRNA and protein levels. Importantly, this reduction occurs whether or not PLD1 is depleted by siRNA (Fig. 6B-F). While the ability of PT and PT-AuNS treatments to rescue transcript/protein/staining levels after PLD1-siRNA treatment is striking, it is not evidence that these extracts work through PLD1, rather it suggests an PLD1-independent mechanism that is strong enough to overcome the PLD1-siRNA phenotype. Therefore, the authors either need significantly more evidence to support their claims that PT extracts work through PLD1, or they require testing an alternate hypothesis for the mechanism by which these extracts affect adipogenesis before this work can be accepted for publication.

Major Issues:

·        PT extract effect through PLD1 not supported by the evidence presented (see above)

·        Why are AuNS without PT-functionalization not used as a control? While it is clear that the PT-AuNS are effective at significantly lower amounts than PT extract by itself, data showing that AuNS by themselves are not driving similar effects is missing (in other words, the authors either need to show that the effects of PT-AuNS are not driven solely by the nanostructures but by the functionalized nanostructures, or provide an explanation as to why this control cannot be performed or is not appropriate).

Minor Issues:

·        The introduction mostly references obesity caused by ‘proliferation and differentiation of preadiposites’. While this may technically correct, it misses the nuance in the adipocyte field that obesity is caused by hypertrophy (fat cells getting bigger) and hyperplasia (or adipocyte proliferation). While the authors do seem to recognize this distinction (mentioning adipocyte number and volume), hypertrophy is not necessarily the same process as ‘differentiation’ and this distinction needs to be clarified.

·        The authors claim that “…the size of adipocytes can be controlled through dietary regulation; however, controlling the differentiation of new precursor adipocytes into adipocytes is ineffective through dietary regulation.” However, they do not cite any literature or data supporting this claim.

Author Response

Thank you very much for allowing us to revise our manuscript entitled, “Characterization of Plocamium telfairiae extract-functionalized Au nanostructures and their anti-adipogenic activity through PLD1 (marinedrugs-1786570)”. We appreciate the reviewers for their constructive comments, which were very helpful for improving our paper. The manuscript has been carefully revised according to the reviewers’ comments. The revisions are marked in red in the revised manuscript. The detailed responses (in BOLD type) to the comments are provided below.

In their manuscript “Characterization of Plocamium telfairiae Extract-Functionalized Au Nanostructures and their Anti-Adipogenic activity through PLD1” Park et al. describe a method for functionalizing Au nanostructures with ethanolic extract from seaweed, and testing this for its anti-adipogenic effects. While this functionalization method is of interest to the field, the author’s conclusion that the anti-adipogenic effects are through PLD1 are not supported by the data presented. The authors show that the PT and PT-AuNS treatments are able to reduce many markers of adiposity in MDI-treated fibroblasts, including Oil Red O and Nile Red staining as well as C/EBPα, PPARγ, SREBP-1, FAS, and aP2 mRNA and protein levels. Importantly, this reduction occurs whether or not PLD1 is depleted by siRNA (Fig. 6B-F). While the ability of PT and PT-AuNS treatments to rescue transcript/protein/staining levels after PLD1-siRNA treatment is striking, it is not evidence that these extracts work through PLD1, rather it suggests an PLD1-independent mechanism that is strong enough to overcome the PLD1-siRNA phenotype. Therefore, the authors either need significantly more evidence to support their claims that PT extracts work through PLD1, or they require testing an alternate hypothesis for the mechanism by which these extracts affect adipogenesis before this work can be accepted for publication.

Major Issues:

  • PT extract effect through PLD1 not supported by the evidence presented (see above)

Response: There are only a few studies of the link between adipogenesis and PLD, and the function and molecular mechanisms of PLD in adipogenesis are unclear. Therefore, our research team is currently conducting a research project to elucidate the role of PLD in adipogenesis and will publish a related paper. To summarize the findings so far, deletion of PLD1 increased adipogenesis, body fat mass, and hepatic steatosis along with upregulation of PPAR-γ and C/EBPa. BAT, WAT, and body fat mass increase were confirmed. It was confirmed that this regulation is made through the Wnt-β-catenin signaling pathway. Based on this result, screening was performed to select PLD1 target anti-obesity candidates, and it was confirmed that PT-AuNS had an indirectly related effect. We investigated whether PT-AuNS directly inhibited PLD1 expression in preadipocytes and mature adipocytes, but experimental results showed that PT-AuNS did not directly inhibit the mRNA and protein levels of PLD1 in preadipocytes and mature adipocytes. However, in the results of FIG. 6 , when comparing the control group with the Si-PLD1 group, it was confirmed that the lipogenesis significantly increased in the si-PLD1 group compared to the control group. It was confirmed that these effects were inhibited to a similar level in PLD1-deficient mature adipocytes and mature adipoctye when PT-AuNS treatment was performed. Therefore, manuscript 2.6. of "Therefore, in PLD1-deficient adipocytes, differentiation is promoted through FAS upregulation via PPAR-γ and C/EBPα induction, and these effects are inhibited by PT extract and PT-AuNS." The text was modified as follows. “Therefore, in mature adipocytes deficient in PLD1, differentiation was more promoted through FAS upregulation through PPAR-γ and C/EBPa induction than in mature adipocytes, and it was confirmed that this effect was indirectly inhibited by PT-AuNS.”

  • Why are AuNS without PT-functionalization not used as a control? While it is clear that the PT-AuNS are effective at significantly lower amounts than PT extract by itself, data showing that AuNS by themselves are not driving similar effects is missing (in other words, the authors either need to show that the effects of PT-AuNS are not driven solely by the nanostructures but by the functionalized nanostructures, or provide an explanation as to why this control cannot be performed or is not appropriate).

Response: Our research team verified the Oil Red O staining assay using AuNS synthesized by citric acid. As a result, it was confirmed that citric acid-AuNS was reduced by 12.4 ± 2.4% at 0.5 mg/ml.

Minor Issues:

  • The introduction mostly references obesity caused by ‘proliferation and differentiation of preadiposites’. While this may technically correct, it misses the nuance in the adipocyte field that obesity is caused by hypertrophy (fat cells getting bigger) and hyperplasia (or adipocyte proliferation). While the authors do seem to recognize this distinction (mentioning adipocyte number and volume), hypertrophy is not necessarily the same process as ‘differentiation’ and this distinction needs to be clarified.

Response: This sentence has been removed in response to a reviewer's comment.

“Obesity at the cellular level is caused by an increase in the number and volume of adipocytes through the promotion of their proliferation and differentiation. Therefore, controlling adipogenesis the process of differentiation from preadipocytes to mature adipocytes is an important step in obesity treatment [9].”

  • The authors claim that “…the size of adipocytes can be controlled through dietary regulation; however, controlling the differentiation of new precursor adipocytes into adipocytes is ineffective through dietary regulation.” However, they do not cite any literature or data supporting this claim.

Response: According to the reviewer's comments, the literature was cited in the relevant sentence.

  1. Murugan, D.D.; Balan, D.; Wong, P.F. Adipogenesis and therapeutic potentials of antiobesogenic phytochemicals: Insights from preclinical studies. Phytother Res 2021, 35, 5936-5960.

Reviewer 2 Report

The manuscript entitled “Characterization of Plocamium telfairiae Extract-Functionalized  Au Nanostructures and their Anti-Adipogenic Activity through  PLD1’’ investigated whether PT-functionalized AuNSs (PT-AuNSs) can alleviate MDI-induced mature adipocyte activation and whether the mechanism underlying their possible anti-adipogenic effects is through the inhibition of phospholipase D1 (PLD1)-mediated adipogenesis. The idea of this work is very interesting and novel. In addition, the used methods, results and discussion parts are written in a professional manner. However, I have some suggestions:

1.     The authors mentioned “Au nanostructure (AuNS) biosynthesis was mediated through an ethanolic extract of Plocamium telfairiae (PT)” the authors used ethanolic extract (polar) why they did not investigate also nonpolar extract e.g. hexane. They have any evidence that the polar extract contains the active constituents.

2.     At the end of the abstract part, kindly write about your recommendations for future use of the biosynthesized particles.

3.     In the introduction part, I suggest adding a paragraph about the WHO's last statistics about obesity.

4.     In the 3.2. Preparation of PT Extract section kindly write who identified Plocamium telfairiae algae, where identified, and also where deposited the sample and voucher specimen code.

5.     Statistical analysis is missing in the conducted work results and the significance of the results must be shown well.

6.     Add references to all used methods.

7.     The manuscript needs some editing, grammar, and typo correction

Author Response

Thank you very much for allowing us to revise our manuscript entitled, “Characterization of Plocamium telfairiae extract-functionalized Au nanostructures and their anti-adipogenic activity through PLD1 (marinedrugs-1786570)”. We appreciate the reviewers for their constructive comments, which were very helpful for improving our paper. The manuscript has been carefully revised according to the reviewers’ comments. The revisions are marked in red in the revised manuscript. The detailed responses (in BOLD type) to the comments are provided below.

The manuscript entitled “Characterization of Plocamium telfairiae Extract-Functionalized  Au Nanostructures and their Anti-Adipogenic Activity through  PLD1’’ investigated whether PT-functionalized AuNSs (PT-AuNSs) can alleviate MDI-induced mature adipocyte activation and whether the mechanism underlying their possible anti-adipogenic effects is through the inhibition of phospholipase D1 (PLD1)-mediated adipogenesis. The idea of this work is very interesting and novel. In addition, the used methods, results and discussion parts are written in a professional manner. However, I have some suggestions:

  1. The authors mentioned “Au nanostructure (AuNS) biosynthesis was mediated through an ethanolic extract of Plocamium telfairiae (PT)” the authors used ethanolic extract (polar) why they did not investigate also nonpolar extract e.g. hexane. They have any evidence that the polar extract contains the active constituents.

Response: Metal nanostructures are used as biosensors and gene-targeted drug delivery systems in the medical field because of their easy surface modification, uniform particle size distribution, and excellent stability. Metal nanostructures vibrate free electron clouds through mutual interference with electromagnetic waves of external light to form surface plasmon resonance and exhibit biocompatibility properties. Specifically, AuNSs can be easily integrated with biological molecules because of their low biotoxicity, and their ability to adsorb various organic ligands to the surface. In the past decade, research has been conducted to devise new eco-friendly and inexpensive synthesis methods that deviate from the existing approaches that toxic chemicals. Plant extracts, plant biomolecules, and biological materials such as bacteria, fungi, and algae are eco-friendly approaches for safe, alternate modes of production.  AuNSs can be synthesized using physical and chemical methods. For physical synthesis, expensive equipment is required, and controlling the size of nanostructures is difficult. In the case of chemical synthesis, the process is simple but expensive and has the side effect of reducing agents. Accordingly, active research is being conducted to devise a method for AuNSs biosynthesis without the use of organic solvents. Therefore, PT ethanolic extract was used as a reducing agent and stabilizer in this study.

Andleeb, A., Andleeb, A., Asghar, S., Zaman, G., Tariq, M., Mehmood, A., Nadeem, M., Hano, C., Lorenzo, J.M., Abbasi, B.H., 2021. A Systematic Review of Biosynthesized Metallic Nanoparticles as a Promising Anti-Cancer-Strategy. Cancers (Basel) 13, 2818. doi: 10.3390/cancers13112818.

Boomi, P., Ganesan, R., Prabu Poorani, G., Jegatheeswaran, S., Balakumar, C., Gurumallesh Prabu, H., Anand, K., Marimuthu Prabhu, N., Jeyakanthan, J., Saravanan, M., 2020. Phyto-Engineered Gold Nanoparticles (AuNPs) with Potential Antibacterial, Antioxidant, and Wound Healing Activities Under in vitro and in vivo Conditions. Int. J. Nanomedicine 15, 7553-7568.

Holišová, V., Urban, M., Konvičková, Z., Kolenčík, M., Mančík, P., Slabotinský, J., Kratošová, G., Plachá, D., 2021. Colloidal stability of phytosynthesised gold nanoparticles and their catalytic effects for nerve agent degradation. Sci. Rep. 11, 4071-021-83460-1.

Jun, E.S., Kim, Y.J., Kim, H.H., Park, S.Y., 2020. Gold Nanoparticles Using Ecklonia stolonifera Protect Human Dermal Fibroblasts from UVA-Induced Senescence through Inhibiting MMP-1 and MMP-3. Mar. Drugs 18, 433. doi: 10.3390/md18090433.

Machado, S., González-Ballesteros, N., Gonçalves, A., Magalhães, L., Sárria Pereira de Passos, M., Rodríguez-Argüelles, M.C., Castro Gomes, A., 2021. Toxicity in vitro and in Zebrafish Embryonic Development of Gold Nanoparticles Biosynthesized Using Cystoseira Macroalgae Extracts. Int. J. Nanomedicine 16, 5017-5036.

Nayem, S.M.A., Sultana, N., Haque, M.A., Miah, B., Hasan, M.M., Islam, T., Hasan, M.M., Awal, A., Uddin, J., Aziz, M.A., Ahammad, A.J.S., 2020. Green Synthesis of Gold and Silver Nanoparticles by Using Amorphophallus paeoniifolius Tuber Extract and Evaluation of Their Antibacterial Activity. Molecules 25, 4773. doi: 10.3390/molecules25204773.

  1. At the end of the abstract part, kindly write about your recommendations for future use of the biosynthesized particles.

Response: The last part of the abstract part has been modified based on the reviewers' advice.

“In this study, the biosynthesized PT-AuNS was used as a potential therapeutic candidate because it conferred a potent anti-lipogenic effect. As a result, it can be used in various scientific fields such as medicine and the environment.”

  1. In the introduction part, I suggest adding a paragraph about the WHO's last statistics about obesity.

Response: The introductory section has been revised according to the advice of reviewers.

“According to WHO statistics for 2021, 13% of adults worldwide are obesity and 39% of adults worldwide are overweight. In particular, 1 in 5 adolescents and children worldwide is overweight.”

  1. Mir, I.A.; Soni, R.; Srivastav, S.K.; Bhavya, I.; Dar, W.Q.; Farooq, M.D.; Chawla, V.; Nadeem, M. Obesity as an Important Marker of the COVID-19 Pandemic. Cureus 2022, 14, e21403.
  1. In the 3.2. Preparation of PT Extract section kindly write who identified Plocamium telfairiae algae, where identified, and also where deposited the sample and voucher specimen code.

Response: The Materials and Methods section has been revised based on the reviewer's advice.

“PT was purchased from Jeju Technopark Biodiversity Research Institute (gift certificate sample number JBRI-16041). PT specimens were stored in the Herbarium of the Jeju Institute of Biological Diversity, and the identification of deposited PTs was performed by Dr. Wookjae Lee (Jeju Technopark, Jeju).”

  1. Statistical analysis is missing in the conducted work results and the significance of the results must be shown well.

Response: The manuscript was revised according to the opinions of the reviewers.

“All analyses were done using SPASS 22.0 software (SPSS Inc., New York, USA), and GraphPad Prism 4.0 0 (GraphPad, San Diego, CA, USA) was used for graphing. Quantitative data were expressed as mean±standard deviation (mean±SD), and P<0.05 was statistically significant. Cell viability, oil-red, Nile-red, Triglyceride rate of adipocyte, the relative expression of C/EBPa, PPARg, SREBP-1, FAS,  aP2, UCP1, PRDM16, and PGC1a  mRNA and proteins among different groups were analyzed using one-way analysis of variance (ANOVA) followed by Tukey’s post-hoc test.”

  1. Add references to all used methods.

Response: According to the reviewer's comments, the literature was cited in the relevant sentence.

  1. The manuscript needs some editing, grammar, and typo correction

Response: The manuscript was revised according to the opinions of the reviewers.

Reviewer 3 Report

The authors investigated the anti-adipogenic effects of ethanolic extracts of Plocamium telfairiae (PT) and PT-functionalized AuNSs (PT-AuNSs) in 3T3-L1 cells. PT-AuNS reduced intracellular triglyceride level with lowered expression of adipogenic C/EBPα, PPARγ, SREBP-1, FAS, and aP2 genes/proteins through PLD1. In addition, PT-AuNS induced the expression of UCP1, PRDM16, and PGC1a, thus indicating induction of browning. However, there is no in-depth research on the cellular and molecular mechanisms involved. PLD1. How does PLD1 regulate expression of adipogenic genes/proteins? Mechanistic analysis is not enough. The roles of PLD1 in this regulation is not clear. There are concerns that should be addressed.

1.        The abbreviated words should be shown by the full-spelling, when they were first appeared.

2.         How many times were the studies replicated? It should be clearly indicated.

3.         The authors focused PLD1 in this regulation. Why did the authors analyzed PLD1-mediated pathway? The roles of PLD1 in the regulation is unclear. Why knockdown of PLD1 decreased adipogenic gene/protein expression. What mechanisms are involved in this regulation.

4.         PLD1 expression in adipocytes is correlated with PPARg and C/EBPa expression in Ref. 46. However, in Ref. 46, such representation was not found. It should be confirmed.

5.         What is mammalian protein extraction reagent? Correct name should be shown.

6.         Overall the description should be extensively revised. Especially, methods should be revised to reproduce the study by the readers. Moreover, the figure legends should be revised. There is no explanation for statistically significant differences.

7.         What amount of proteins were loaded in each lane in Western blot analysis?

8.         There is no information about PLD1-specific siRNA. Moreover, when siRNA was transfected during adipogenesis. When changed the medium, siRNA was transfected? It should be clearly mentioned.

Author Response

Thank you very much for allowing us to revise our manuscript entitled, “Characterization of Plocamium telfairiae extract-functionalized Au nanostructures and their anti-adipogenic activity through PLD1 (marinedrugs-1786570)”. We appreciate the reviewers for their constructive comments, which were very helpful for improving our paper. The manuscript has been carefully revised according to the reviewers’ comments. The revisions are marked in red in the revised manuscript. The detailed responses (in BOLD type) to the comments are provided below.

The authors investigated the anti-adipogenic effects of ethanolic extracts of Plocamium telfairiae (PT) and PT-functionalized AuNSs (PT-AuNSs) in 3T3-L1 cells. PT-AuNS reduced intracellular triglyceride level with lowered expression of adipogenic C/EBPα, PPARγ, SREBP-1, FAS, and aP2 genes/proteins through PLD1. In addition, PT-AuNS induced the expression of UCP1, PRDM16, and PGC1a, thus indicating induction of browning. However, there is no in-depth research on the cellular and molecular mechanisms involved. PLD1. How does PLD1 regulate expression of adipogenic genes/proteins? Mechanistic analysis is not enough. The roles of PLD1 in this regulation is not clear. There are concerns that should be addressed.

  1. The abbreviated words should be shown by the full-spelling, when they were first appeared.

Response: The manuscript was revised according to the opinions of the reviewers.

  1. How many times were the studies replicated? It should be clearly indicated.

Response: Figure legends have been revised according to the opinions of the reviewers.

  1. The authors focused PLD1 in this regulation. Why did the authors analyzed PLD1-mediated pathway? The roles of PLD1 in the regulation is unclear. Why knockdown of PLD1 decreased adipogenic gene/protein expression. What mechanisms are involved in this regulation.

Response: There are only a few studies of the link between adipogenesis and PLD, and the function and molecular mechanisms of PLD in adipogenesis are unclear. Therefore, our research team is currently conducting a research project to elucidate the role of PLD in adipogenesis and will publish a related paper. To summarize the findings so far, deletion of PLD1 increased adipogenesis, body fat mass, and hepatic steatosis along with upregulation of PPAR-γ and C/EBPa. BAT, WAT, and body fat mass increase were confirmed. It was confirmed that this regulation is made through the Wnt-β-catenin signaling pathway. Based on this result, screening was performed to select PLD1 target anti-obesity candidates, and it was confirmed that PT-AuNS had an indirectly related effect. We investigated whether PT-AuNS directly inhibited PLD1 expression in preadipocytes and mature adipocytes, but experimental results showed that PT-AuNS did not directly inhibit the mRNA and protein levels of PLD1 in preadipocytes and mature adipocytes. However, in the results of FIG. 6 , when comparing the control group with the Si-PLD1 group, it was confirmed that the lipogenesis significantly increased in the si-PLD1 group compared to the control group. It was confirmed that these effects were inhibited to a similar level in PLD1-deficient mature adipocytes and mature adipoctye when PT-AuNS treatment was performed. Therefore, manuscript 2.6. of "Therefore, in PLD1-deficient adipocytes, differentiation is promoted through FAS upregulation via PPAR-γ and C/EBPα induction, and these effects are inhibited by PT extract and PT-AuNS." The text was modified as follows. “Therefore, in mature adipocytes deficient in PLD1, differentiation was more promoted through FAS upregulation through PPAR-γ and C/EBPa induction than in mature adipocytes, and it was confirmed that this effect was indirectly inhibited by PT-AuNS.”

  1. PLD1 expression in adipocytes is correlated with PPARg and C/EBPa expression in Ref. 46. However, in Ref. 46, such representation was not found. It should be confirmed.

Response: The text has been corrected in the manuscript based on the reviewer's comments.

  1. What is mammalian protein extraction reagent? Correct name should be shown.

Response: The text has been corrected in the manuscript based on the reviewer's comments.

“M-PER™ Mammalian Protein Extraction Reagent”

  1. Overall the description should be extensively revised. Especially, methods should be revised to reproduce the study by the readers. Moreover, the figure legends should be revised. There is no explanation for statistically significant differences.

Response: The manuscript was revised according to the opinions of the reviewers.

“All analyses were done using SPASS 22.0 software (SPSS Inc., New York, USA), and GraphPad Prism 4.0 0 (GraphPad, San Diego, CA, USA) was used for graphing. Quantitative data were expressed as mean±standard deviation (mean±SD), and P<0.05 was statistically significant. Cell viability, oil-red, Nile-red, Triglyceride rate of adipocyte, the relative expression of C/EBPa, PPARg, SREBP-1, FAS,  aP2, UCP1, PRDM16, and PGC1a  mRNA and proteins among different groups were analyzed using one-way analysis of variance (ANOVA) followed by Tukey’s post-hoc test.”

  1. What amount of proteins were loaded in each lane in Western blot analysis?

Response: The manuscript was revised according to the opinions of the reviewers.

“Adipocytes were lysed using a M-PER™ Mammalian Protein Extraction Reagent containing a protease inhibitor. The mixture was placed on ice for 30 min, shaken every few minutes, and centrifuged at 12,000 rpm and 4°C for 10 min. The supernatant was collected, and protein concentration was measured using a Bio-Rad protein assay kit (Bio-Rad Laboratories, Inc.). MiniPROTEAN Precast Gels (Bio-Rad Laboratories, Inc.) were prepared, and the samples (30 g) were loaded following protein denaturation. A Hybond polyvinylidene difluoride membrane (Amersham, Cytiva) was selected, and the proteins were transferred onto the membrane. The membranes were blocked for 1 h and incubated overnight at 4°C with diluted primary antibodies. The primary antibodies used are listed in Table 2. The membranes were then washed three times and incubated with horseradish peroxidase-conjugated secondary antibody for 1 h at room temperature. After washing, the membranes were incubated with enhanced Pierce ECL western blotting substrate. The relative intensities of bands were determined using ImageQuant TL (version 8.1; Amersham, Cytiva).”

  1. There is no information about PLD1-specific siRNA. Moreover, when siRNA was transfected during adipogenesis. When changed the medium, siRNA was transfected? It should be clearly mentioned.

Response: The manuscript was revised according to the opinions of the reviewers.

“PLD1-specific siRNA, and negative control siRNA were purchased from Thermo Fisher Scientific Life Sciences.

Adipocytes were transfected with mouse PLD1-specific siRNA and negative control siRNA using X-treme GENE siRNA transfection reagent according to the manufacturer's instructions. The X-treme GENE siRNA transfection reagent, control siRNA, and si-PLD1 were mixed with Opti-MEM I Reduced Serum Medium and then activated for 20 min at room temperature, transfection mixture with control siRNA, and si-PLD1 were then added to the adipocytes. After 24 h of incubation, fresh medium was added and the adipocyte were grown for additional 48 h.  Gene silencing was confirmed using RT-qPCR. To confirm the anti-adipogenic effect, PT-AuNS (10–20 mg mL-1) was administered for 9 days 2 h before exposure to MDI differentiation medium.”

Round 2

Reviewer 1 Report

In their response, the authors failed to address my two major concerns. First, they summarize additional experiments for a separate paper linking PLD1 to adipogenesis. While this is certainly an important avenue of research, it does not address the underlying concern, that the authors did not provide evidence for their claim that PT or PT-AuNS work through PLD1. Their revised statement Therefore, in mature adipocytes deficient in PLD1, differentiation was more promoted through FAS upregulation through PPAR-γ and C/EBPa induction than in mature adipocytes, and it was confirmed that this effect was indirectly inhibited by PT-AuNS” is insufficient for two reasons. 1) the authors did not provide evidence that adipocyte differentiation was “more promoted through FAS upregulation through PPARγ and C/EBPa…”; all they showed in this manuscript, and are therefore able to support, is that the mRNA and protein levels of these proteins increased due to PLD1 knockdown. In the absence of a published paper on their additional research in this area, the authors either need to provide data to support their statements, or alter the statements to reflect the data shown (ex. In mature adipocytes deficient in PLD1, FAS, PPARγ, and C/EBPa transcripts and protein levels were increased…). 2) while the second part of that statement is acceptable and supported by the data (“…it was confirmed that this effect was…inhibited by PT-AuNS”), it does not show that the effect of PT or PT-AuNS is mechanistically through PLD1 as claimed in the manuscript title and conclusion. If the authors altered their claims to “PT and PT-AuNS can mitigate or rescue PLD1-knockdown related increases in adiposity” that statement is supported by the data and would be acceptable.

Second, in their response the authors claim that citric acid-derived Au-NS reduced Oil red O staining by ~12% (without showing data). This experiment is an important control and should be incorporated into the manuscript (it was not) with appropriate statistics (probably as part of Fig 2), especially given that a significant part of the effect of PT-AuNS (~12.5 to 15% of the ~80% reduction in Oil Red O staining) can be attributed just to the AuNS, and appears to be comparable to the reduction by PT treatment alone. Is the 12% reduction with citric acid-AuNS statistically significantly different than control? Than PT-AuNS? Additionally, how are the transcript and/or protein levels of FAS, PPARγ, and C/EBPa affected by citric acid-AuNS (Fig. 4)? Mitochondrial thermal production (Fig. 5)? Do they show a similar 12% reduction to the Oil Red O staining? These questions need to be addresses before the manuscript is suitable for publication.

Author Response

Thank you very much for allowing us to revise our manuscript entitled, “Characterization of Plocamium telfairiae extract-functionalized Au nanostructures and their anti-adipogenic activity through PLD1 (marinedrugs-1786570)”. We appreciate the reviewers for their constructive comments, which were very helpful for improving our paper. The manuscript has been carefully revised according to the reviewers’ comments. The revisions are marked in red in the revised manuscript. The detailed responses (in BOLD type) to the comments are provided below.

In their response, the authors failed to address my two major concerns. First, they summarize additional experiments for a separate paper linking PLD1 to adipogenesis. While this is certainly an important avenue of research, it does not address the underlying concern, that the authors did not provide evidence for their claim that PT or PT-AuNS work through PLD1. Their revised statement Therefore, in mature adipocytes deficient in PLD1, differentiation was more promoted through FAS upregulation through PPAR-γ and C/EBPa induction than in mature adipocytes, and it was confirmed that this effect was indirectly inhibited by PT-AuNS” is insufficient for two reasons. 1) the authors did not provide evidence that adipocyte differentiation was “more promoted through FAS upregulation through PPARγ and C/EBPa…”; all they showed in this manuscript, and are therefore able to support, is that the mRNA and protein levels of these proteins increased due to PLD1 knockdown. In the absence of a published paper on their additional research in this area, the authors either need to provide data to support their statements, or alter the statements to reflect the data shown (ex. In mature adipocytes deficient in PLD1, FAS, PPARγ, and C/EBPa transcripts and protein levels were increased…). 2) while the second part of that statement is acceptable and supported by the data (“…it was confirmed that this effect was…inhibited by PT-AuNS”), it does not show that the effect of PT or PT-AuNS is mechanistically through PLD1 as claimed in the manuscript title and conclusion. If the authors altered their claims to “PT and PT-AuNS can mitigate or rescue PLD1-knockdown related increases in adiposity” that statement is supported by the data and would be acceptable.

Response: The manuscript (Result and Disccusion 2.6. and Figure 6A) was revised according to the opinions of the reviewers.

“To determine whether PLD1 knockdown is required for induction of adipogenic differen-tiation, we transfected pre- and mature adipocytes with PLD1-specific siRNA and nega-tive control siRNA. Interestingly, adipogenic differentiation of PLD1-specific siR-NA-transfected mature adipocytes was significantly increased by mRNA levels of C/EBPa, PPARγ and FAS (Figure. 6a). To determine whether PLD1 could mediate the PT extract- and PT-AuNS-induced anti-adipogenesis, adipocytes were transfected with mouse PLD1-specific siRNA or negative control siRNA and then induced with MDI differentia-tion medium. As shown in Figure 6b, PLD1 transcript expression was knocked down through PLD1 siRNA transfection. We investigated whether PLD1 deficiency upregulated the expression of C/EBPα, PPARγ, and FAS. Indeed, the transcript and protein expression levels of C/EBPα, PPARγ, and FAS were increased in the PLD1 siRNA-transfected group, and these effects were inhibited by PT extract and PT-AuNS (Figure 6c, d, and e). In addi-tion, Nile red staining was performed to investigate whether PLD1 expression is required for lipid droplet formation. Lipid droplet formation increased due to the lack of PLD1, and the size of droplets formed was also increased. These observations were further confirmed by the suppressive effects of PT extract and PT-AuNS (Figure 6f and g). Therefore, in ma-ture adipocytes deficient in PLD1, differentiation was more promoted through FAS upreg-ulation through PPAR-γ and C/EBPa induction than in mature adipocytes, and it was confirmed that this effect was indirectly inhibited by PT-AuNS. These results suggest that PT-AuNS can mitigate or rescue PLD1-knockdown related increases in adiposity.”

Second, in their response the authors claim that citric acid-derived Au-NS reduced Oil red O staining by ~12% (without showing data). This experiment is an important control and should be incorporated into the manuscript (it was not) with appropriate statistics (probably as part of Fig 2), especially given that a significant part of the effect of PT-AuNS (~12.5 to 15% of the ~80% reduction in Oil Red O staining) can be attributed just to the AuNS, and appears to be comparable to the reduction by PT treatment alone. Is the 12% reduction with citric acid-AuNS statistically significantly different than control? Than PT-AuNS? Additionally, how are the transcript and/or protein levels of FAS, PPARγ, and C/EBPa affected by citric acid-AuNS (Fig. 4)? Mitochondrial thermal production (Fig. 5)? Do they show a similar 12% reduction to the Oil Red O staining? These questions need to be addresses before the manuscript is suitable for publication.

Response: The manuscript (Result and Disccusion 2.2., 2.3 and Figure 2C, D and E, Figure 3) was revised according to the opinions of the reviewers.

We previously performed Oil-red and Nile-red assays for Citrate-AuNS at 20 mg/ml, the same concentration condition of PT-AuNS, and at this concentration, Citrate-AuNS was compared to MDI-induced Oil-red and MDI-induced Oil-red assays and Nile-red activity was not inhibited. Therefore, as a result of an additional experiment by increasing the concentration, a slight decrease was confirmed from 0.5 mg/ml to 12.4 ± 2.4%. Citrate-AuNS did not change the oil-red and Nile-red activity induced by MDI under the same concentration and conditions as PT-AuNS (20 mg/ml), so the subsequent transcript and/or protein levels of FAS, PPARγ, and C/EBPa, etc., were not performed.

Reviewer 2 Report

The authors established all the required corrections, and I have no more comments.

Author Response

We really appreciate your comments.

Reviewer 3 Report

The manuscript was improved.

However, there are still some minor concerns as below.

1. C/EBPα should be unified. “C/EBPα”, “C/EBP α”, and “C/EBP a” were found. It should be confirmed.

2. The sequence of PLD1-specific siRNA should be shown.

Author Response

Thank you very much for allowing us to revise our manuscript entitled, “Characterization of Plocamium telfairiae extract-functionalized Au nanostructures and their anti-adipogenic activity through PLD1 (marinedrugs-1786570)”. We appreciate the reviewers for their constructive comments, which were very helpful for improving our paper. The manuscript has been carefully revised according to the reviewers’ comments. The revisions are marked in red in the revised manuscript. The detailed responses (in BOLD type) to the comments are provided below.

The manuscript was improved.

However, there are still some minor concerns as below.

  1. C/EBPα should be unified. “C/EBPα”, “C/EBP α”, and “C/EBP a” were found. It should be confirmed.

Response: The manuscript was revised according to the opinions of the reviewers.

  1. The sequence of PLD1-specific siRNA should be shown.

Response:  “PLD1-specific siRNA, and negative control siRNA were purchased from Thermo Fisher Scientific Life Sciences. Unfortunately, the exact sequence is not provided for commercial siRNAs.